# Evidence-based action plan for integrating artificial intelligence in an academic medical centre-a multidisciplinary approach

Lena Jafri[1]*, Hafsa Majid[1], Aysha Habib Khan[1], Aqueel Kapadia[1], Shanzay Rehman[1], Kiran Hilal[2], Imran Siddiqui[1], Arsala Farooqui[3], Kulsoom Ghias[4], Naveed Rashid[5], Yousra Sarfaraz[1], Muhammad Umer Naeem Effendi[1], Syed Murtaza Raza Kazmi[6], Muneeb Khalid[7], Khairulnissa Ajani[8], Qamar Riaz[9], Mehmood Riaz[5], Rozina Roshan[10], Sibtain Ahmed[1]

1 Section of Chemical Pathology, Department of Pathology and Laboratory Medicine Aga Khan University (AKU), Karachi Pakistan, 2 Department of Radiology, AKU, Pakistan, 3 Institute for Global Health and Development, AKU, 4 Department of Biological & Biomedical Sciences, AKU, Pakistan, 5 Department of Medicine, AKU, Pakistan, 6 Department of Surgery, AKU, Pakistan, 7 Aga Khan Medical College, Pakistan, 8 School of Nursing & Midwifery, AKU, Pakistan, 9 Department of Post Graduate Medical Education, AKU, Pakistan, 10 Quality and Patient Safety Department and Department of Infection Prevention & Hospital Epidemiology, AKU, Pakistan

* lena.jafri@aku.edu

## Abstract

### Introduction

As the effectiveness of artificial intelligence (AI) in enhancing various facets of healthcare delivery becomes more apparent, it is anticipated that AI will soon find its way into standard clinical practices, even in low and middle-income countries. The objective of this study was to create an action plan for integrating AI into medical education, research, and clinical practices utilizing the SWITCH model of change integrating both rational and emotional aspects.

### Methods

This exploratory qualitative study employed reflexive thematic analysis of semi-structured interviews and a co-design workshop, followed by the collaborative development of an action plan. The study was conducted from May 2023 to May 2024, at the Aga Khan University, Karachi, Pakistan. The development of an action plan was informed by interviews, co-design workshop, and discussions with diverse group of academic leaders, healthcare professionals and medical students. All interviews, workshop sessions, and planning meetings were audio recorded, transcribed verbatim, and anonymized. Data management was conducted manually using Microsoft Word and Excel. Findings after thematic analysis of the qualitative interviews and findings from the co-design workshop were gradually, and inductively transformed into the content for the action plan for integrating AI into healthcare.

**Data availability statement:** All relevant data are within the manuscript.

**Funding:** The author(s) received no specific funding for this work.

**Competing interests:** The authors have declared that no competing interests exist.

**Abbreviations:** AI: Artificial intelligence; AKU: Aga Khan University; AKUH: Aga Khan University Hospital; LMIC: low middle income countries; PGME: Postgraduate Medical Education; UGME: Undergraduate Medical Education; QTL_Net: Quality Network of Quality, Teaching, and Learning; CPD: continuous professional development; EHR: Electronic Health Record; NLP: Natural Language Processing; CPOE: Computerized Physician Order Entry; TLA: Total Lab Automation; CIME: Center of Innovation and Medical Education; VLE: Virtual Learning Environment.

## Results

The content analysis of interviews identified following four themes: AI opportunities, apprehensions toward AI-induced changes, pushing change through leadership styles, and importance of AI related capacity building. During the workshop, participants discussed aligning current AI knowledge with future requirements by identifying clear instructions, emotional motivators, and environmental changes required on the path of AI integration. The proposed action plan conceptualized AI integration as multidimensional change process comprising three domains: strategic actions, change pathway enablement and environmental readiness and human motivation operationalized through twenty actionable components.

## Conclusion

The study findings provide a context specific conceptual action plan for healthcare professionals to integrate AI into medical education, clinical service and research, Future work should focus on pilot implementation and empirical validations of the action plan across diverse healthcare settings to assess feasibility, effectiveness and scalability.

## Introduction

The healthcare system is undergoing a surge of disruptive innovations, including robotic surgery, 3D printing, wearable health devices, personalized medicine, telemedicine, and Artificial Intelligence (AI) integration [1,2]. AI-driven technologies can transform academic medical centers by analyzing clinical data, allowing healthcare providers to tailor management plans to individual characteristics. This approach has the potential to improve patient outcomes and optimization of resources, making it a necessary change especially for low middle income countries (LMIC) [3]. Academic leaders can create an action plan to effectively handle the changes caused by AI integration [4].

Several change management theories focus on the inner, extrinsic and social support factors as facilitators of the change process. Kurt Lewin's three-stage model, 'A systems model of change', 'the theory of change', 'the ADKAR model', and 'Kotter's eight-step approach' all offer ways for supporting change, from preparation to embedding change in organizational culture [5,6]. These models of change primarily emphasize structure and processes, with diminished attention to emotional and cognitive transformation. The SWITCH model of change, developed by Heath and Heath, simplifies dealing with change resistance by emphasizing positive features, goal clarity, and environmental and emotional support. This framework integrates both rational and emotional aspects, making it applicable in various contexts, from personal change to large-scale organizational transformations [7]. Because AI fosters change through a granular participatory approach, Lewin's approach is frequently too simplistic. It sees change as a condition that may be attained and then frozen. The

process of managing change for the integration of AI in healthcare facilities is ongoing. Scalable AI adoption requires that the SWITCH methodology change the emphasis from a destination to a method of working. As for the Kotter's change model, it emphasizes a top-down approach and the creation of urgency. In situations where staff feel overwhelmed by ongoing innovations, increasing urgency can be counterproductive, resulting in employee burnout. The SWITCH model of change emphasizes on strengthening the human resource to address the emotional challenges of AI change, providing a psychological safety net that is often overlooked in top-down models, contrasting with Kotter's vision centric approach. The ADKAR model omits the shape the path aspect. Knowledge alone is insufficient for this type of change; if the data is unstructured or the tools are complex. SWITCH tackles the issue by making the new behavior the path of least resistance. Additionally, the SWITCH model of change promotes rapid testing and adaption, which makes it perfect for integrating AI in healthcare settings where actual change at the individual and team level is required rather than merely creating a high-level organizational change plan. Because of its practical, behavior-focused, and flexible approach, the SWITCH model of change was selected for this study above other theories of change.

Significant efforts have been made to develop AI solutions for healthcare improvement, yet healthcare professionals, particularly in LMICs, face challenges in implementing AI in their daily practice. Existing literature primarily emphasizes the technical performance, validation and accuracy of AI tools, while less focus is given to their integration into clinical service workflows. Change frameworks are often applied retrospectively or theoretically, lacking educationally informed and partic-ipatory processes. Most existing literature on AI in healthcare is conceptual or policy-oriented mostly from the developed countries. Pakistan's AI ecosystem in healthcare has distinct structural, political, infrastructural, and socio-cultural con-straints that are under-represented in the literature. How AI is experienced, negotiated, and implemented by healthcare professionals in developing countries remains insufficiently understood.

This study used a multidisciplinary exploratory educational engagement strategy to support AI integration, where findings from earlier phases informed subsequent activities. Although each activity involved different participants, the sequential learning process allowed insights to accumulate and guide the development of ways of integrating AI in health-care. The objective of the current study was to develop an actionable plan for integration of AI within an academic medical center [the Aga Khan University (AKU) and Aga Khan University Hospital (AKUH) Pakistan] using insights from interviews and workshop, structured according to SWITCH model of change. AKUH is a tertiary care hospital affiliated with academic entities (the Medical College and School of Nursing and Midwifery), and an Academic Medical Centre in Pakistan with an IQRA-based vision that prioritizes Impact, Quality, Relevance, and Access to health care [8]. It is known for its culture of innovation and leadership, as well as its commitment to consistently improving patient care standards and involving the community through outreach programs to raise health awareness and reduce disease burden [9].

## Methods

### Study design and phases

This exploratory qualitative study employed reflexive thematic analysis of semi-structured interviews and a co-design workshop, followed by the collaborative development of an action plan. The current study was conducted in three phases. Phase 1 involved individual semi-structured interviews with institutional leaders and faculty. Phase 2 consisted of a co-design workshop with multidisciplinary participants, utilizing the SWITCH framework as a sensitizing lens. Phase 2 focused on the collaborative development of a SWITCH-aligned action plan, synthesized from the findings of Phases 1 and 2.

### Ethical considerations

Before commencement of the study permission from Aga Khan University's Ethical Review Committee (ERC number 2023-8372-24124) was sought. Written informed consent was taken from all participants. Data collection was completed within the ERC-approved period (before March 2024).

 

## Participants and data collection

The study was conducted from May 2023 to May 2024. Subsequent data cleaning and statistical analysis were conducted from March 2024 onward. Data were collected from various sources and participants via interviews, an interactive co-design workshop, and discussions, ensuring informed consent was obtained from all participants. The composition of participants varied across these Purposive sampling was done to gather data from various educational leaders [associate deans, departmental chairs, section heads, education leads of Postgraduate Medical Education (PGME) and Undergraduate Medical Education (UGME) and multidisciplinary faculty, clinicians] and UGME students, PGME students at AKU and AKUH [10]. The study's objectives and the absence of conflicts of interest were communicated to all the participants. Considering the study's objective, the sample of interest, the theoretical perspectives of the study, and the purposive sample selection, a sample of 10–12 participants was estimated for the interviews. However, data collection through interviews continued until thematic saturation was achieved. Workshop included 22 participants with almost negligible overlap with interview participants. Subsequent meetings for action plan development primarily involved the research team, and authors to refine and validate emerging insights.

Prior to the interviews, a pilot study was conducted with two participants to evaluate the interview schedule's effectiveness, leading to refinements in language and style. Interviewees were contacted through invitation emails and follow-up phone calls to schedule interviews. Semi-structured interviews, lasting 20–30 minutes, were conducted by trained research colleagues with qualitative research experience who also participated in pilot interviews prior to data collection. The research team (n = 4) included both men and women who used interview schedules, based on literature and with clear prompts, to carry out the process. The researchers framed the interviews as discussion sessions rather than performance evaluations to encourage open dialogue and reduce participant anxiety. Face-to-face interviews were audio recorded and thematically analyzed by the research team and the principal investigator. Clarifications were sought during and post-interviews to ensure accuracy, with key findings reviewed in team meetings.

Using interview findings, a workshop aimed to discuss the actionable plan for integrating AI into healthcare. The workshop gathered collaborative data focused on how the SWITCH model of change can be utilized to create a comprehensive framework for AI integration in medical education and in-service practices at an academic medical center. A qualitative validation approach was implemented during the workshop, where participants discussed preliminary themes and findings, offering feedback. To mitigate social desirability bias, confidentiality of all participants was ensured, discussions were held neutrally, and open-ended questions were utilized to foster honesty. Researchers also considered their own biases and experiences to limit their impact on participants, while data triangulation was employed to enhance credibility. This process facilitated the triangulation of perspectives, ensuring that the findings accurately reflected the participants' viewpoints.

An action plan was developed and subjected to rigorous review through closed meetings, allowing for collaborator feedback and refinement to enhance effectiveness and feasibility. The plan was reviewed among authors via meetings and email. All interviews, workshop sessions, and planning meetings were audio recorded, transcribed verbatim, and anonymized. Data management was conducted manually using Microsoft Word and Excel.

## Data analysis

Reflexive thematic analysis was conducted using Braun and Clarke's six-step framework with slight modifications to suit the objective of the study [11], While steps 1–4 followed a systematic, interview based thematic analysis, steps 5 and 6 transitioned into participatory validation through a workshop and the subsequent development of an action plan for change. Microsoft Word and Excel were used to organize and analyze the notes and transcripts from interviews and workshop. As described in Fig 1, the analysis began with an immersion phase, during which research team reviewed the transcripts several times to achieve thorough familiarity with the data. During this step, preliminary observations and reflexive notes were documented to identify initial patterns relevant to the study objectives. As step two of the process,

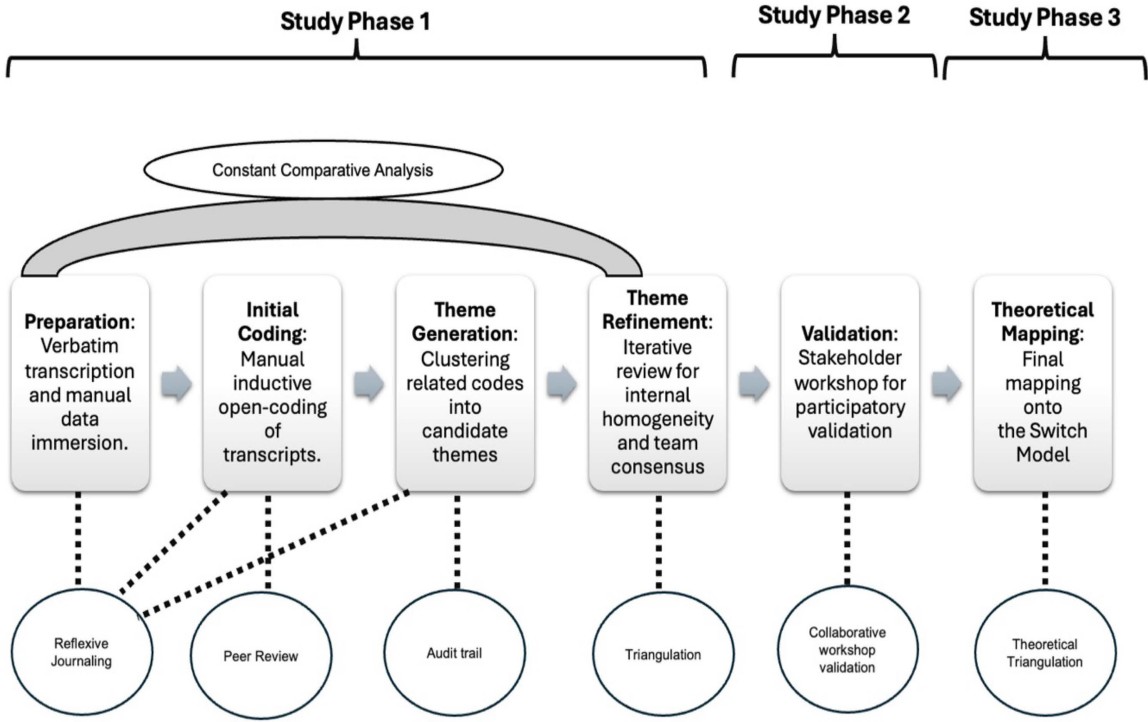

**Fig 1. Conceptual framework for ai integration in academic medical centers: Study phases and thematic analysis with integrated quality check measures.**

the transcripts were divided into smaller units and initially coded by two researchers. Segments of data were labelled to capture meaningful patterns. The team held regular discussions to address coding discrepancies and presented data with illustrative quotes from participants, with their informed written consent. To ensure rigor in this process, the research team collaborated and resolved discrepancies through consensus, thereby enhancing inter-coder agreement. As a third step the codes were grouped into higher level categories to form overarching themes, with the research team reviewing the process for recurring patterns and key concepts. An inductive approach allowed themes to emerge from the data while acknowledging the researchers' interpretative role. The research team mitigated potential bias by grounding the synthesis in participant accounts and reflective coding notes. All records of coding decisions, modifications and rationales were documented. As fourth step, the themes were iteratively reviewed, raw data was revisited and refined to ensure internal homogeneity (all data within a specific theme supported a unified interpretation) and external heterogeneity (ensuring themes were distinct). In the fifth step of the process, specific themes were established and further refined during a multidisciplinary workshop. This refinement aimed to ensure that the themes were both coherent and distinct from one another. Additionally, the workshop participants compared the insights and findings generated during their discussions with the themes identified in the interviews, ensuring a comprehensive evaluation of the data collected from multiple sources. Triangulation was achieved by comparing interview insights with data collected during workshops, capturing multiple perspectives and ensuring consistency across different settings. The use of collaborative problem solving and anonymity further validated the findings by reducing bias and confirming results directly with participants. Finally, as last step, thematic mapping and reporting were conducted, where the validated themes were aligned with the SWITCH model of change. The analytic integration utilized data source triangulation to synthesize interviews, workshop, and meetings into a cohesive action plan. To mitigate potential power imbalances, participants

were guaranteed anonymity, ensuring that individual identities remained disconnected from the data. Potential bias in the analysis phase was managed through data source triangulation and the use of reflective notes to bracket pre-existing assumptions. By reconciling individual interview themes with broader workshop discourse and institutional planning data, the team ensured the final SWITCH-based framework was grounded in collective evidence rather than researchers led narratives. Findings were compared for thematic complementarity and subsequently reconciled through a rigorous process based on viability, stakeholder relevance, and institutional alignment. This iterative validation process ensured that the final SWITCH based action plan was not merely a thematic summary, but a strategically calibrated system designed to shape the path of AI change while addressing the documented elephant resistance of the workforce. By categorizing the emergent themes under the domains of directing the rider, motivating the elephant, and shaping the path, the analysis moved from descriptive findings to a structured framework for guiding AI related institutional change.

## Results

### Interview findings

A total of eighteen interviews were conducted, representing professionals from diverse departments including Pathology and Laboratory Medicine, Women and Child Health, Dean's Office, Hospital Operations, Department for Educational Development, Emergency Medicine, Pediatrics, Community Health Sciences, Surgery, School of Nursing, Quality Network of Quality, Teaching, and Learning (QTL_Net) and the Medical College of AKU. The male-to-female ratio among the participants was approximately 1.25 to 1. The range of years of experience among the professionals varied from 9–32 years, with a mean experience of $14.5 \pm 5.1$ years. Content analysis revealed following overarching themes: the potential of AI opportunities, resistance toward AI-induced changes, maximizing learning development and resources in AI to drive transformation through leadership styles, and importance of trainings/ capacity building to sustain change (Table 1).

### AI opportunities for improvement

The interviewees framed AI not only as a technological upgrade of healthcare systems but as a performance enabling method for healthcare professionals. Predictive analytics and integrated AI-powered diagnostics were frequently mentioned by participants as becoming standard in clinical workflows to address labor shortages and improving efficiency. Several participants expressed enthusiasm about integrating AI into their administrative responsibilities to streamline hospital operations including providing care to patients at the bedside. Furthermore, responses from participants illustrated a wider awareness among healthcare professionals that AI solutions could be used for patient interaction, data flow, and patient management in addition to acting as channels for service delivery. This could indicate a change in hospital operations and professional duties within the healthcare system by shifting responsibility from hospital workers to digital interfaces. The AI tools were viewed as facilitators of timely and informed clinical decision making through rapid access to patient data and decision support systems. As one respondent (P16) noted, *"AI finds utilization in triage in emergency medicine for better patient outcomes"* highlighting the role of AI in strengthening patient flow and time management in emergency settings. Participants envisioned an integrated healthcare system enabling patients to connect handheld devices to hospital networks for instant transmission and analysis of medical data, promoting individual care and overcoming cognitive limits. One of the participants (P12) remarked, *"integrated healthcare system may not only enhance convenience for patients but also ensure quicker and more accurate decision-making by healthcare professionals, ultimately leading to improved patient outcomes."* Participants accounts suggested that AI use shifts the clinical decision making from reliance on individual judgement alone to a more data supported and system assisted process. A significant tension emerged between the initial expectations of AI as a productivity facilitator and the emergent risks of clinical dependence. Some also expressed apprehensions that increased dependence on AI may lead to over reliance to automated systems

**Table 1. Leadership insights from structured interviews: themes, categories, and codes.**

| Themes or Overarching Ideas | Patterned Meaning or Categories | Indicative Codes or Data Labels |
|---|---|---|
| Opportunities for improvement | Perceived benefits of AI anchored in present day clinical practice | Knowledgebase of AI in the institute |
| | | How they position themselves in relation to AI related changes |
| | | Status of AI application in the institute |
| | | Availability of AI-tools |
| | Optimistic imaginaries to implement AI at their healthcare institution | Advantages of AI in research |
| | | Advantages of AI usage in service |
| | | Advantages of AI usage in education |
| | | Limitations of current healthcare practices |
| | | Improved patients' outcome |
| Reluctance to Embrace Change | Personnel risk of AI integration related to roles, responsibilities and accountability | Receptibility of stakeholders |
| | | Job insecurity, professional uncertainty and overreliance on AI over healthcare professionals |
| | | Knowledge gaps |
| | | Traceability of wrong clinical decisions |
| | | Challenges faced in adopting AI by users, strategic fragmentation |
| | | Ethical concerns |
| | Lack of connect between AI ambitions in the institute and availability of technical, educational and financial resources | Limitations of current AI-tools |
| | | Financial challenges in approving use of AI |
| | | Supply challenges in approving use of AI |
| | | Lack of AI experts |
| Enhancing Learning and Growth | Active willingness in identifying educational gaps and providing resources to build AI competence for effective practice | Educational reforms needed identifying AI as a core professional skill |
| | | Resources for teaching and learning about the utility of AI in the institute for AI upskilling |
| | | Hands on courses and research to enhance AI skills to improve organizational readiness to embrace this change |
| Leadership Styles and Practices | Need for formal guidance and consistent AI integration strategies | Suggestions on how to implement AI in practice |
| | | Identified need and readiness to develop concrete actionable plan to adopt AI |
| | Collective organizational strategies to compensate for limited resources | AI advocates and AI enthusiasts required |
| | | Interdepartmental collaboration and resource sharing |
| | AI Leadership to ensure safe AI practices | Visionary leaders, unified strategic goals |
| | | Overcoming change obstacles |
| | | Continuous assessment to sustain change management |

raising concerns about bias in clinical decision making. While participants anticipated that AI would alleviate cognitive load, they simultaneously expressed a drift toward automation bias, where over reliance on algorithmic outputs threatened to undermine individual clinical accountability and lead to the deskilling of the medical workforce.

Participants further indicated that benefits of AI use could extend across many clinical departments. They associated AI related transformation with improvements in diagnostics across many disciplines. As one respondent (P2) explained "*AI is poised to advance in diagnostics, leveraging image analysis in pathology, radiology, enabled by rapid digitization of glass slides, mass spectrometry chromatograms and digital imaging technology*" suggesting that integration of AI and diagnostic technologies are perceived as important drivers for accurate and timely patient management. One of the respondents (P5) associated AI related advancements is reshaping expectations in diagnostic laboratories, *"in a laboratory setting, natural intelligence is utilized for analyzing organism colony morphology, biochemical reactions, microscopy, and susceptibility patterns based on previous experiences. my expectation with AI is going to be much higher than what I deliver right now and what my seasoned technologists deliver right now."* The participants responses indicated that they expect AI to handle tedious tasks, complex clinical data exceeding current performance standards thus raising their expectations significantly compared to current human capabilities.

Across the interviews, participants highlighted the potential of AI tools to personalize learning pathways for medical students automate grading tasks with real-time feedback, paper setting and exams scoring. Additional uses include allowing students to set their own schedules and advance at different rates. This implied a more comprehensive rethinking of physicians as learning facilitators, interpreters of AI-generated insights, and not just content providers. As AI takes over the role of content provider by allowing students to advance at their own rates, the physician's role may shift toward learning facilitation and the interpretation of AI generated insights. This represents a sociotechnical tension where the opportunity for student autonomy creates an existential challenge for clinicians, who need to redefine their professional value beyond their traditional expertise.

## Reluctance to embrace change

The healthcare providers' accounts of challenges linked to AI integration in their hospital highlighted both structural and psychological dimensions of organizational change. The participants consistently identified that any attempt at implementing AI would be met with resistance due to the fear of losing their job. Some participants expressed that AI being a new subspecialty would create jobs, but a significant hurdle would be acceptability. Concerns regarding job security highlight the tension between AI as an innovation and its potential threat to professional identity, revealing occupational anxiety and uncertainty about workforce stability and responsibilities in hospitals. Some participants framed AI as an enhancer of professional skills while maintaining job security, viewing its inclusion in healthcare systems as a complement to their roles rather than a substitute. As one participant (P14) explained, *"AI was never meant to and is never meant to replace people"* highlighting the continued importance of judgement in clinical care.

Participants also acknowledged the challenge of introducing a major organizational change, emphasizing the need for meticulous planning to effectively adopt AI technology. This necessitates ongoing communication to articulate needs, expectations, and seek expert guidance for AI implementation. While the institute's checks and balances ensure organizational safety and integrity, they can also impede or halt beneficial changes. Concerns regarding data privacy, patient consent, and cybersecurity were highlighted as both regulatory and moral obligations in the context of AI integration in hospitals. They were hesitant to fully endorse AI as a tool in medical education due to unresolved ethical issues and inadequate regulatory oversight. Participants' acceptance coexisted with mild reservations, indicating that organizational, psychological, and ethical concerns had just as big of an impact on AI adoption as functionality and apprehension about employing AI technologies in healthcare. Participants identified the import of AI technology as a significant barrier to change, particular in a country like Pakistan where reliance on imported technology increases procurement complexity and cost. As a result, AI integration is a resource-intensive shift in which the availability of AI tools, cost, and infrastructure are critical to driving AI integration into healthcare. The participants highlighted concerns regarding reliance on imported AI solutions noting limited local development capacity as a potential barrier to sustainable implementation. As one respondent (P3) stated, *"all our lab consumables even Eppendorf are imported, implying that AI tools are costly to import, and*

*overall, AI integration costs make it difficult to sustain. If Pakistan had its own local AI systems for hospitals, we wouldn't worry about import costs and integration would be sustainable."* Locally developed AI systems could help overcome barriers associated with high cost and import dependency. Interviewees expressed concerns regarding the importance of ensuring that imported AI solutions are matched to local needs, are validated on clinical data and conform to all relevant regulations. As one respondent (P9) noted, *"I have seen lots of AI algorithms lack the evidence needed to support implementation. Clinicians will have to demand substantial resources either from the institute or the vendor to validate these tools in our local population"* highlighting the need for local validations, extra work and time and stronger support from leadership. Another participant (P2) stated *"AI makes diagnostics faster but makes the doctor's day slower because of the validation burden"*. The findings reveal a contradictory drift where the opportunity for efficiency was not viewed as a simple benefit, but as a catalyst for anxiety regarding job security and the potential de-skilling of the clinical workforce. In the domain of diagnostics, a sharp contradiction emerged between the technical speed of AI tools and the temporal reality of clinical practice. While participants identified the opportunity for quick diagnostic interpretations, this was overshadowed by a validation burden. This created a drift from efficiency to increased workload, as clinicians reported that the cognitive effort of auditing an uninterpretable AI would exceed the time required for a traditional manual diagnosis. This matters because it suggests that AI does not simply save time but rather reconfigures clinical workload, shifting the burden from task execution to high-stakes oversight, which can lead to increased mental fatigue and burnout.

## Leadership styles and practices

Participants emphasized that the extent of AI integration hinges upon the vision of its leaders. Leaders possess the ability to empower and facilitate AI-related change within the institution. To achieve this, they themselves must first acquire a comprehensive understanding of AI-based healthcare solutions and subsequently endeavor to maximize the benefits while mitigating associated challenges. Effective leadership entails advocating for change in a manner that alleviates fears and encourages acceptance of AI technology among others. Strong leadership to handle this change with robust IT security, are essential for the successful integration of AI in healthcare stated another respondent (P13), *"Strong AI leaders in our hospital can reframe its use from just speed and efficiency to accountability and reduction in clinical errors"*. The role of AI integration in hospitals as a real-time auditing mechanism was discussed. One of the respondents (P9) stated, "leadership should clearly communicate that AI transition is non-negotiable and as a tool for patient safety and transparency" thereby emphasizing that the transition towards AI systems enhances leadership oversight and transparency through the availability of traceable audit trails and real time monitoring of clinical activities.

Leadership for AI integration in hospitals is strategic and relational rather than just administrative. The capacity of the leader to incorporate AI into the current workflow, cultivate trust, and reaffirm the crucial role of human health care workers in addition to AI tools will determine if the transition is effective. Participants varied opinions regarding leader's approach to change management. Some preferred a more authoritative stance, while others advocated for a collaborative approach, involving users and stakeholders in planning and execution. Some also rationalized that AI-leadership can also emerge from individuals at lower levels who are passionate about the change and committed to supporting and guiding others. Like one participant (P11) stated*, "a tech-savvy AI leader must be found in the institution, who does not have to be a departmental or institutional head and may be anyone as long as they are knowledgeable and enthusiastic in driving this change,"* suggesting that expertise and AI competence may play a more important role than hierarchical position in leading such a massive project. A significant hierarchical drift emerged, where participants identified a shift from top-down management to distributed leadership. Technical proficiency, often found among junior staff or frontline clinicians, superseded traditional seniority. This creates a sociotechnical tension where the expert is no longer defined by years of clinical experience, but by their ability to navigate and implement AI tools, effectively decentralizing power within the hospital structure. Regardless of the approach, participants stressed that team building and strategic planning were crucial components for effective AI integration. Putting together committed cross-functional teams with people with a range of leadership

philosophies, operational expertise, and clinical and AI understanding is essential. Forming a team is an effective organizational strategy to integrate AI practices, promote group ownership, and moderate possible resistance to change.

## Capacity building of healthcare providers and students

Based on the interview data, limited infrastructure and shortage of AI-specialized professionals were the areas identified for attention. Participants consistently underlined that several training exercises prior to implementation were necessary to acquaint stakeholders and everyone else with AI to reduce errors, foster trust and confidence and clarify shifting roles brought by AI tools. According to the responses, the success of integrating AI into healthcare would also depend on trained personnel. It is crucial to have instructors or hospital officials who are knowledgeable about AI technology as role models and to instruct peers and pupils. Besides trainings, concerns about ethical issues prompt calls for regulatory policies, procedures and rigorous validation of AI tools to ensure reliability, patient safety and efficacy prior to use. Once formulated all involved (medical teachers and students) must be aware and practice these ethics in AI related regulations and policies. Validation of AI tools was framed not only as technical prerequisite but a confidence building mechanism to allowing healthcare providers to use AI tools with reduced fear of errors. Participants identified how AI integration can be a threat to patient care if healthcare providers are replaced with non-validated AI tools. One participant (P14) stated that *"validating AI algorithms is crucial. Many overlook its importance and ethical implications for patients"* highlighting concerns about accountability and responsible implementation. Similarly, another participant (P5) emphasized that *"AI integration can jeopardize patient care if we gatekeepers are replaced with non-validated automated solutions"* underscoring the importance of clinical validation and regulatory oversight prior to clinical adoption. Hence, the staff must be aware of AI tools validation processes and policies before implementing it. A significant thematic drift emerged within capacity building, where the focus expanded from individual technical proficiency to institutional policy literacy. Participants identified building capacity as a process of understanding the regulatory and ethical boundaries of AI use, shifting the clinician's role from a simple end-user to a policy aware practitioner. This creates a tension between the desire for rapid innovation and the necessity for procedural compliance, where the speed of AI adoption is naturally constrained by the workload of policy validation. Participants assessed financial resources as necessary not just for acquiring AI technologies and an effective integrated clinical database, but also for sustaining training validation and system maintenance, in addition to cyber protection. Participants noted that AI algorithms require clean, curated clinical data. The Electronic Health Record (EHR) offers hope by centralizing medical records and data, improving accessibility and efficiency. Participants identified gaps in technical expertise and infrastructure as barriers to AI integration in healthcare. As one of the participants (P4) stated; "there *is a deficiency of expertise, resources, and finances. Achieving progress requires investment in human resources and systems like EHR,"* reflecting concerns of readiness of institutions to support this change.

## Workshop findings

A workshop to strategize AI implementation in healthcare was held at AKU in collaboration with the Quality Network of Quality, Teaching, and Learning (QTL_Net) AKU. A diverse group of 22 individuals, including research analysts, faculty, PhD students, and undergraduate and postgraduate medical students from various departments (information technology, internal medicine, oncology, pathology, radiology, and psychiatry) participated in the workshop. Table 2 describes the collected data from this workshop concerning the identification of SWITCH components.

## The action plan for institutional change

An analytical framework that captures how healthcare providers think about, negotiate, and react to AI in their current situations was developed from participant narratives. The proposed action plan conceptualized AI integration as multidimensional change process comprising of three domains: strategic actions for clinicians and healthcare providers (Direct the Rider), change pathway enablement and motivation and attitude (Motivate the Elephant) and environmental readiness

**Table 2. SWITCH components for change management to integrate AI in education, service and research in a hospital identified by workshop participants.**

| Direct the Rider | Motivate the Elephant | Shape the Path |
|---|---|---|
| **Find Bright Spots**<br>• Seeking guidance from local and international AI experts<br>• Find successful AI projects locally<br>• Identify departments where AI adoption led to improved outcome<br>• Showcase AI-projects.<br>• Recognize AI-enthusiasts with impact | **Find the Feeling**<br>• AI Advocacy<br>• Engaging leadership.<br>• Justification for AI investments<br>• Financial feasibility of AI projects<br>• Regular evaluation of AI projects and action plan | **Tweak the Environment**<br>• Collaborate on AI-initiatives.<br>• Allocating resources for AI-infrastructure.<br>• Align with organizational goals |
| **Script Critical Moves**<br>• Integrating AI concepts into the curriculum<br>• Prototyping AI solutions<br>• Establishing an ethical AI Policy across education, research, and service | **Shrink Change**<br>• AI Tools Availability<br>• Provide AI skills<br>• Adress resistance to change AI related clinical practices | **Build Habits**<br>• AI training programs<br>• Integrate AI into workflows<br>• Encourage reflection on AI usage<br>• Establish accountability for AI practices<br>• Recognize AI enthusiasts |
| **Point to Destination**<br>• Prioritizing AI applications.<br>• Setting goal of implementing AI within the healthcare setting<br>• Validate AI algorithms locally | **Grow Your People**<br>• AI taskforce/ committee to lead the change<br>• Encouraging continuous learning related to AI<br>• Encourage AI innovations.<br>• AI Rewards sAI Awards | **Rally the Herd**<br>• Leadership engagement & commitment<br>• Communicate AI action plan<br>• AI trainings at all levels<br>• AI community building<br>• Identify AI change agents.<br>• Celebrate small wins |

The Rider represents the analytical and logical side of thinking, focusing on planning and execution. The Elephant represents the emotional and instinctive side of thinking, focusing on motivation and engagement. Each of these points addresses aspects of motivation, engagement, and emotional buy-in necessary for successful change management. the "Path" represents the environmental or situational factors that influence behavior change. It involves tweaking the environment to make the desired behavior easier and more likely to occur.

(Shape the Path). The resulting action plan represents an analytic synthesis of the qualitative themes identified in Phase I. The action plan represents an analytic synthesis of the primary study themes, including opportunities for improvement, reluctance to change, leadership styles, and capacity building. By mapping these emergent findings onto the SWITCH framework, the plan moves beyond a generic policy guideline to provide a contextually grounded roadmap for institutional change. The proposed action plan for incorporating AI into research, education, and services at an academic healthcare facility is as follows:

**Direct the rider**

1. **Engagement with AI Experts:** To direct the rider, the action plan includes engagement with AI experts to provide the technical clarity and strategic guidance identified as necessary during the interviews. This action is an analytic synthesis of the capacity building theme, addressing the knowledge gaps reported by participants regarding AI integration and ethical governance. The need for multidisciplinary collaborations and the engagement of AI experts is highlighted by the fact that clinical expertise alone is insufficient to address the technical, ethical, and infrastructural aspects of AI-healthcare systems. AI experts can offer important guidance and expertise in implementing AI solutions, assisting in the successful navigation of complex technologies and methodologies. Collaborating with interdisciplinary teams, local

and external AI experts, and individuals with experience ensures the accuracy, reliability, and safety of AI-driven healthcare solutions, leveraging diverse perspectives and resources for successful implementation.

2. **Curriculum Integration**: The analysis highlights how a thorough medical curriculum (undergraduate and graduate) that covers both theoretical knowledge and real-world applications in healthcare settings must incorporate AI concepts and skills. To direct the rider, the plan proposes curricular reforms as a primary strategy for capacity building. This action is an analytic synthesis of participant concerns regarding the current lack of technical readiness. Specialized AI-related courses and continuous professional development (CPD) for physicians, supervisors, and teachers were identified as crucial enablers in enhancing both foundational knowledge and practical skills in the application of AI in healthcare. Capacity building and training in essential AI concepts such as machine learning, deep learning, and neural networks are emphasized as fundamental to the integration of AI in healthcare. These concepts support various applications including diagnostics, personalized medicine, predictive analytics, data management, medical image analysis, and clinical decision support systems. By integrating AI competencies into the formal curriculum, the framework ensures that future healthcare professionals possess the foundational knowledge required for ethical and effective AI utilization.

3. **Ethical, Legal, and Regulatory Considerations**: By providing concrete legal guidelines, the plan gives the 'Rider' the technical clarity needed to proceed with AI adoption without the barrier of ambiguity. This is an analytic synthesis of the institutional challenges identified by participants, specifically the uncertainty regarding data privacy and liability. The action plan emphasizes the importance of ethics as a critical domain in the planning process for bringing AI in healthcare. It highlights the necessity to discuss and address ethical issues, including patient privacy, data security, algorithmic bias, and accountability. The study reveals a significant gap between the concerns surrounding AI in healthcare and the actual needs of healthcare providers. It highlights the necessity for an inclusive policy framework aimed at standardizing operations and addressing the potential risks noted by study participants. Such a policy should focus on protecting both patients and healthcare professionals by promoting the responsible advancement and application of AI technologies, while also safeguarding privacy, autonomy, and data security. Adherence to ethical standards will also help uphold equity in healthcare delivery and comply with legal and regulatory requirements.

4. **Strategic Prioritization:** The findings highlight a sequential approach to prioritizing AI applications within organizations. This includes assessing their impact, scalability, feasibility, and alignment with organizational goals. Additionally, fostering a supportive AI culture and focusing on outcomes rather than processes in strategic planning are emphasized as crucial elements of this approach. To operationalize this component there is value in leveraging existing AI platforms in the institute like Computerized Physician Order Entry (CPOE), AKU's Center of Innovation and Medical Education (CIME), Total Laboratory Automation (TLA), Virtual Learning Environment (VLE) and the ongoing initiative of AKU of Electronic Health Record systems. This approach not only minimize redundancy but capitalize on staff familiarity. One such example is AI-supported simulators in CIME which are in current times enabling professionals to safely practice procedures using real-world scenarios, utilizing AI algorithms for training purposes. The integration of CIME with VLE can be used as a medium to certify and re-certify for credentialling of faculty and staff to provide safe care to the patients. The VLE can also be strategically strengthened to provide clinicians with access to teaching and learning resources, case studies, and examples of clinical best practices. These examples are presented as illustrative options, showing possibilities or approaches. This approach directly addresses the capacity building theme by providing the clear roadmap that institutional leaders often lack. By dividing projects into quick wins (high-impact, low-effort) and long-term goals, the framework gives staff the specific, measurable targets needed to track progress and stay accountable.

5. **Objective Setting and Measurement:** Importance of clear evidence-based objectives ensures alignment between strategy and execution of AI integration in the healthcare institutes. Specifying measurable objectives along with key

performance indicators (KPI) is crucial for the successful implementation of AI, as it allows for the development of realistic timelines and key milestones. Furthermore, it establishes clear metrics or KPIs for evaluating performance effectively and will strengthen governance. This is an analytic synthesis of the institutional challenges and capacity building themes, addressing the lack of measurable objectives found in the data. By defining specific technical and clinical targets, the framework provides the rider with the necessary clarity to evaluate success and maintain accountability.

6. **Pilot Programs and Prototyping:** The action plan incorporates pilot projects as an essential stage for feasibility assessment. This is an analytic synthesis of the institutional challenges and capacity building themes, as it allows for the iterative testing of protocols and data collection instruments in a controlled setting. By identifying technical flaws or workflow deficiencies early, the pilot provides the 'Rider' with the empirical evidence needed to refine the plan before full-scale implementation. These pilot studies are intended to evaluate viability, gather feedback, and refine implementation plans. Suggested actions include integrating AI into existing initiatives like EHR, CPOE, and AI-supported simulators in CIME and VLE to enhance healthcare outcomes through targeted projects.

**Motivate the Elephant.**

1. **Advocacy and Leadership Engagement:** To motivate the elephant, the action plan emphasizes 'making noise' through consistent internal communication and the celebration of small wins. Analysis suggests that the engagement and support of local leadership play a critical role in motivating the staff and in sustainable AI integration in the institute. Leaders may analyze the potential impact of establishing a database of AI experts and enthusiasts on supporting and advancing AI projects within their institute Potential leverage points for coordinated efforts and collaboration among AI project participants are suggested by the idea of holding regular casual open mic events or meetups to promote conversations on AI projects. This directly addresses the reluctance to change identified in the qualitative findings. By increasing the visibility of early successes, the plan counters emotional resistance and builds a sense of institutional momentum and shared excitement, transforming AI from a perceived threat into a collective opportunity.

2. **Accessibility of AI Materials:** Analysis indicated that the initiative of AI integration in healthcare is influenced by availability and accessibility of AI-related resources, including tools, software, textbooks, online courses, journals, workshops, research papers, and educational tools. Ensuring the immediate availability of user-friendly AI tools serves to motivate the elephant by reducing the intimidation factor often associated with new technology. this directly addresses the reluctance to change identified in the interview findings. This will reduce cognitive load, encourage curiosity and creativity, and fuel intrinsic motivation to explore AI concepts. Analysis of potential resource avenues to provide opportunities to the healthcare providers would be required. These may include local and global AI funding opportunities, government AI-initiatives, multi- disciplinary or multi-institute collaborative AI projects and educational AI projects which can be identified and shared across the institute. When tools are accessible and produce quick, helpful results, it builds professional confidence, shifting the perception of AI from an added burden to a valuable supportive partner.

3. **Cost-Benefit Analyses:** Conducting thorough cost-benefit analyses is crucial for justifying AI investments and securing funding from stakeholders. Performing a cost-benefit evaluation serves to motivate the elephant by transforming AI from a financial risk into a strategic advantage. This comprehensive evaluation helps identify and mitigate potential risks, reassures stakeholders, and builds confidence in the investment decision. The tangible benefits of AI initiatives, such as improved patient outcomes or cost savings, are more likely to attract investment. Clearly communicating benefits like increased revenue and improved patient outcomes builds institutional confidence and mitigates the emotional fears related to budget reallocations and resistance to change.

4. **Change Management Strategies:** For the effective sustainability of changes related to AI within healthcare facilities, it is crucial to maintain active communication with staff members. Transformation strategies are essential for motivation

 

by creating a culture of psychological safety and shared vision. This involves not only articulating the vision for AI transformation but also sharing feedback, as well as presenting examples of both successful and unsuccessful AI implementations. Such transparency fosters an environment of trust and collaboration. The data indicate that managing resistance to AI integration in an institution, requires a comprehensive approach involving communication, understanding their concerns, engaging them in co-designing solutions, providing trainings to build trust and confidence. This includes transparent explanations of AI implementation's benefits, regular updates, open forums, and motivational open mic sessions. This approach directly addresses the reluctance to change identified in the interviews. By using inclusive decision-making, the strategy transforms AI from an imposed requirement into a collective institutional mission, building the emotional buy in needed for long-term sustainability.

5. **Continuous Learning and Innovation:** Based on the interviews, the identified reluctance to change stems from a perceived threat to professional identity and job security. To counter this, the institutes can adopt a continuous learning strategy. This approach includes giving the staff the time and resources to evolve their skills incrementally, keeping them informed about the latest AI developments in healthcare, applications and encouraging staff to share insights and ideas for AI innovation.

6. **Evaluation and Iterative Improvement:** By shifting the focus to incremental progress, the institute can aim to settle the elephant's anxiety regarding job displacement or technical inadequacy. Regularly evaluate the overall impact of AI in medical education and service delivery. However, To ensure the continuous learning strategy remains effective and the staff emotions remains motivated, project evaluations must shift from pass/fail inspections to developmental milestones. Regular evaluations can be conducted as coaching conversations rather than top-down reporting.

7. **Celebrate small wins:** Celebrating small wins serves as a direct psychological solution to the specific barriers identified in the interviews. Institute leadership may consider acknowledging AI innovations, AI initiatives, individuals performing well with AI tools and take put time to celebrate their efforts. For sustaining motivation and momentum, small wins and successful AI projects can be recognized through regular AI rewards and awards. Successful AI projects with tangible impacts can be linked to ongoing professional evaluations and performance appraisals. This strategy builds psychological safety by rewarding the effort of continuous learning, ensuring staff feel supported rather than pressured during the transition.

**Shape the path.**

1. **Capacity Building:** A critical insight from the interviews was that many individuals felt "not competent enough" to meet new AI related demands. By providing targeted training and resources, the framework builds the emotional confidence and technical skill needed to shape the path of the staff, transforming "incompetence" into a new professional identity. The approach aims to enhance AI literacy and competency among stakeholders through comprehensive capacity-building initiatives, including workshops, seminars, and online resources. It provides education and training programs on AI concepts, tools, and applications, empowers staff at all levels, and offers hands-on experience with AI tools. Regular practice and experimentation reinforce learning and proficiency. Change agents within the organization are identified to champion AI adoption and drive change. The action plan shapes the path by converting identified barriers into environmental tweaks such as simplified processes or better support tools that make the right actions easier to perform. This structural alignment ensures that capacity building isn't just a standalone activity, but it clears the path and empowers the people to move along it sustainably.

2. **Collaborative Learning Spaces:** As data analysis has indicated, the successful implementation of AI in hospitals requires support from clinical staff, and collaborative development can positively influence the AI related transformation. The action plan operationalizes collaborative learning spaces as a structural intervention to mitigate the barriers

identified during the interview phase, specifically the perceived lack of individual competence. In the SWITCH model, this shift fosters a sense of community ensuring that AI implementation is supported by mutual exchange rather than isolated effort. By connecting to the concept of visionary leadership, these environments serve as crucial venues for interdisciplinary problem-solving, where leaders focus on encouraging the generation of ideas rather than simply giving orders.

3. **Infrastructure and Resource Allocation:** Common factors such as structured organizational medical health records, financial resources, and technical infrastructure may serve as foundational elements for effective AI integration in healthcare systems. It's essential to assess and allocate resources for infrastructure upgrades, data management, and user-friendly AI tool acquisition, ensuring the environment is effectively supported for effective AI implementation by planning and assessing existing infrastructure needs. This alignment ensures that the action plan does not merely mandate change but provides the structural machinery that renders the desired strategic outcome the path of least resistance. Data analysis also revealed how an overabundance of AI tools and applications may make it challenging to use and maintain, especially when there isn't enough capacity to handle everything efficiently. Therefore, ensure that AI infrastructure and resource selection is guided by strategic needs and local context. By embedding these technical assets into the organizational topography, the action plan mitigates the barriers of cognitive overload and data fragmentation that contributed to the baseline perception of individual incompetence.

4. **Organizational Alignment:** The action plan establishes organizational alignment as the foundational catalyst for institutional change, directly addressing the interview subthemes of "strategic fragmentation" and "professional uncertainty". The hospital board's support for technologically advanced AI systems and the integration of AI in hospitals is one financial and political enabler. If the AI-related process changes and AI change management approach are important to the entire plan and support the hospital's overarching objectives, they will be feasible and effective. Involve stakeholders and executives in AI projects to encourage creativity and alignment with corporate objectives. Prioritize the engagement of a variety of stakeholders in the planning and decision-making processes. Such engagement and active communication allow for nuanced understanding of local contexts, makes incorporation of AI into workflows simple with ownership, and enhances acceptability and adaptability.

5. **Risk Management and Compliance**: As AI introduces speed and scale in healthcare, necessitating oversight from healthcare professionals to bridge the gap between AI and risk management as well as accountability. Human oversight is vital to respect healthcare professionals' autonomy and minimize adverse consequences. Risk management provides the psychological safety needed to overcome the instinct to refuse change as discussed by the interviewees and workshop participants. By establishing clear ethical policies for AI application and legal compliance structures, the framework transforms AI from a perceived threat into a managed tool. This directly countering the sub-theme of "fear of the unknown" identified in the interviews. Developing robust risk management and compliance strategies is crucial to navigate the ethical, legal, and regulatory challenges associated with AI in healthcare. It is imperative to conduct comprehensive risk mitigation measures for each patient-involved AI project before implementation.

6. **Continuous Monitoring and Feedback:** The action plan utilizes iterative feedback loops as a primary mechanism to shape the path of change, addressing the interview theme of barriers by replacing ambiguity with real time navigational clarity. The feedback channels function as essential psychological scaffolding reducing the anxiety of the unknown and fostering the confidence required for sustained AI adoption. Data analysis further suggests establishing accountability mechanisms and providing support structures to encourage commitment to AI integration. Recognizing and rewarding successful AI adoption can incentivize continued engagement and motivate others to adopt similar habits. For sustainability of AI related change in the healthcare facility active communication with the staff along with feedback, sharing of AI-success stories and AI-failures is essential. This continuous exchange of insights not only directs the rider toward

more efficient critical moves but also rallies the herd by transforming individual successes into shared organizational knowledge, thereby ensuring the change remains both scalable and culturally embedded.

7. **Sustainability and Scalability:** Healthcare regulatory and compliance controls are crucial for the effective deployment of AI in healthcare. Data analysis highlights the importance of ongoing education to enhance AI skills, ensuring successful integration and sustainability within healthcare sectors. Strategies for the sustainability and scalability of AI initiatives ought to focus on capacity building, knowledge transfer, long-term funding, and continuous stakeholder communication to address concerns and maintain enthusiasm. As identified in the interviews the leadership plays a key role by aligning technology with clinical practice, promoting ethical use, and securing resources to foster trust. Sustainability and scalability are achieved by embedding capacity building into the organizational culture, ensuring that the initial bottom-up successes of leaders become reproducible habits. By addressing the barriers of incompetence through a collaborative learning space, the action plan creates a self-sustaining environment where AI implementation can scale naturally across interdisciplinary teams.

## Discussion

Interviews at a Pakistani academic medical center uncovered significant themes regarding the integration of AI in healthcare, emphasizing opportunities, resistance to change, leadership practices, and the importance of building AI-related capabilities. These findings were further explored in a workshop that aimed to incorporate the SWITCH change management model. This study enhances the existing literature on AI in healthcare by proposing an action plan that utilizes the SWITCH model to align strategic direction, execution capability, and behavioral readiness in AI operations. The SWITCH model action plan introduces a process-oriented approach, showcasing key factors influencing AI adoption. It elaborates on the interplay of critical factors, including behavioral, motivational, and change-management dimensions, alongside technological readiness, throughout the AI implementation process in academic medical centers. It moves beyond conceptual prescriptions by empirically grounding AI implementation within the everyday practices of healthcare stakeholders, a dimension underexplored in prior conceptual frameworks.

The study collected insights from healthcare personnel through various methods, highlighting the dual role of AI as a progress driver and a source of concern. It also involved consultations with AI experts and addressed ethical, legal, and regulatory aspects to ensure safe and reliable AI solutions. A triangulated methodology, utilizing interviews and workshops, facilitated a comprehensive understanding, aiming to refine the action plan for successful AI integration in medical education and practice [10,11]. The action plan focuses on addressing the human and organizational facets of AI implementation in healthcare, recognizing that such implementation is still in its early stages and requires further exploration. A scoping review published in 2022 identified research on AI implementation frameworks in healthcare, focusing on publications since 2000 [12]. Out of 2541 screened articles, only seven met the eligibility criteria; two provided formal frameworks for AI implementation. New domains highlighted in the review included data reliance, existing processes, shared decision-making, human oversight, and ethical considerations concerning population impact and inequality. Although previous frameworks have identified barriers, enablers, and evaluation steps, they frequently do not provide comprehensive integration across various perspectives [13–21]. Rahim et al synthesized existing studies to identify enablers and barriers and mapped these to a three-horizon roadmap to guide hospitals toward becoming learning health systems with successful AI integration. This study complements existing systematic review by Rahim et al on AI-enabled learning health systems by translating identified enablers and barriers into a stakeholder-informed action plan [22]. While Nene and Hewitt propose a framework for implementing AI in South African public hospitals, grounded in existing literature and contextual constraints, our study extends this work through an empirical, stakeholder-driven approach translating conceptual considerations into actionable and learning-oriented practices [23]. Notably, the development of structured, stakeholder-informed AI frameworks is essential not only in resource-constrained environments but also in technologically advanced health systems. For

example, digitally advanced health systems such as Australia are actively developing AI governance frameworks [24]. The registered protocol plans to combine a scoping literature review, document analysis, stakeholder interviews, and validation workshops to produce a governance model intended for a major Australian public health organization with multiple hospital sites highlighting the ongoing complexity and evolution of responsible AI implementation as a challenge. The practical, structured checklist FAIR-AI developed in United States, demonstrates that AI implementation should be treated like a clinical intervention and is a continuous clinical governance process not a one-time technical deployment [25]. While patient safety, audits of AI tools and their outcomes were identified as paramount in our study, FAIR-AI also detailed the need for an AI framework to evaluate the impact of potential solutions on health system. The FAIR-AI reinforces the need for learning-oriented, stakeholder-informed approaches like AI action plan developed in this study.

In LMICs like Pakistan AI is accelerating but many hospitals are still moving cautiously. Multiple study participants cited how managerial skills and leadership insights are essential for bridging the gaps between AI and clinical practice and to expedite the change. As identified by current study and reported in published reviews, research studies and conceptual literature on AI, there is a need for robust managerial, ethical, and quality systems to ensure successful AI applications [26–31]. Healthcare regulatory and compliance controls must ensure AI-enabled solutions are safe and effective and promote a culture where AI failures are learned from [32]. Emotions may be conveyed throughout this implementation phase, so leaders must remain connected [33]. The challenges associated with AI related changes are manageable with effective communication from the beginning and beyond [34]. The action plan for integrating AI into healthcare demonstrates significant potential for improving operational efficiency and patient outcomes, setting a model for other institutions [35,36].

This study has certain limitations that should be considered:

- The current study was conducted in a single private hospital in Pakistan hence the findings may not be generalizable to other hospitals with different challenges and digital infrastructure.

- In this era of fast-moving AI advancements, the study may not reflect changing attitudes and perception of healthcare providers as we gathered data at one point in time.

- The data from the interviews mainly included leadership who may already be inclined towards this AI related transformation. To address this potential selection bias, we have tried to include participation from multidisciplinary faculty and various levels of students ensuring broader perspectives and balanced representation.

- We acknowledge the potential for insider bias given the lead researcher's role in data collection and analysis. This study's principal investigator along with the research team maintained dual roles as both investigators and active members of the studied institution. This proximity to the subject matter introduces the risk of social desirability bias; participants may have altered their responses either consciously or unconsciously to align with perceived institutional expectations or to maintain professional rapport with the researchers. The research team's familiarity with the organization may have led them to overlook routine issues (blind spots) that an external observer would have identified as significant. To mitigate this, we tried to maintain analytical distance through reflexive self-review (principal investigator maintained a reflective diary throughout the study) and ensure that findings are grounded in the participants' data. While being 'insiders' helped the research team understand the organization deeply, it may have affected the study's objectivity. Furthermore, the use of an inductive analytic approach allowed for uncomfortable themes like the perceived lack of competence and fear of change to emerge organically, rather than being filtered out by institutional loyalty. Therefore, the findings represent a shared perspective of the institution rather than a completely neutral view.

- The reliance on the SWITCH framework may have biased the results by overlooking perspectives outside of that model. Additionally, because there was no external review or pilot testing of the final plan, its success and usefulness in different contexts are not yet proven. These factors limit the current credibility and applicability of the study's

conclusions. Consequently, these results should be viewed as an initial developmental stage. For the successful adaptation of the proposed action plan by other hospitals, local validation of its effectiveness is essential. It is crucial for these institutions to establish clear performance indicators, encompassing audit mechanisms and outcome measures. Furthermore, the implementation of healthcare AI in Pakistan should prioritize resource availability, capacity building, relevance to local clinical needs, and ethical governance. Future work should therefore focus on pilot implementation and empirical validation of the action plan across diverse healthcare settings to assess its feasibility, effectiveness, and scalability.

## Conclusion

The study outlines a comprehensive strategy for integrating AI into healthcare, focusing on strategic change management, AI competency enhancement, effective leadership, continuous education, and ethical frameworks. It helps healthcare professionals understand the complexities of AI adoption. AI has the potential to improve healthcare delivery, patient outcomes, and address challenges in LMIC hospitals; however, it must be ethically developed, culturally sensitive, and aligned with LMIC healthcare needs in order to be sustainable.

## Acknowledgments

Since the corresponding author was the recipient of the Aga Khan University's QTL_Network's Award for Collaborative Practice Scholarship of Teaching and Learning SoTL, the study was carried out with support from QTL_Net.

## Author contributions

**Conceptualization:** Lena Jafri, Hafsa Majid, Arsala Farooqui.

**Data curation:** Lena Jafri, Hafsa Majid, Aqueel Kapadia, Arsala Farooqui, Sibtain Ahmed.

**Formal analysis:** Lena Jafri, Hafsa Majid, Aqueel Kapadia, Yousra Sarfaraz, Muhammad Umer Naeem, Sibtain Ahmed.

**Funding acquisition:** Aysha Habib Khan, Kulsoom Ghias.

**Investigation:** Lena Jafri, Hafsa Majid, Aqueel Kapadia, Shanzay Rehman, Kiran Hilal, Imran Siddiqui, Arsala Farooqui, Naveed Rashid, Muneeb Khalid, Khairulnissa Ajani, Qamar Riaz, Mehmood Riaz, Rozina Roshan.

**Methodology:** Lena Jafri, Hafsa Majid, Aysha Habib Khan, Kiran Hilal, Kulsoom Ghias, Naveed Rashid, Yousra Sarfaraz, Muhammad Umer Naeem, Syed Murtaza Raza Kazmi, Muneeb Khalid, Khairulnissa Ajani, Qamar Riaz, Mehmood Riaz, Rozina Roshan, Sibtain Ahmed.

**Project administration:** Lena Jafri, Hafsa Majid, Shanzay Rehman.

**Resources:** Lena Jafri, Hafsa Majid, Shanzay Rehman, Imran Siddiqui, Arsala Farooqui.

**Supervision:** Lena Jafri, Hafsa Majid, Imran Siddiqui.

**Validation:** Lena Jafri, Hafsa Majid, Aqueel Kapadia, Shanzay Rehman, Aysha Habib Khan, Kiran Hilal, Kulsoom Ghias, Naveed Rashid, Yousra Sarfaraz, Muhammad Umer Naeem, Syed Murtaza Raza Kazmi, Muneeb Khalid, Khairulnissa Ajani, Qamar Riaz, Mehmood Riaz, Rozina Roshan, Sibtain Ahmed.

**Visualization:** Lena Jafri, Hafsa Majid.

**Writing – original draft:** Lena Jafri, Aqueel Kapadia.

**Writing – review & editing:** Hafsa Majid, Shanzay Rehman, Aysha Habib Khan, Kiran Hilal, Imran Siddiqui, Arsala Farooqui, Kulsoom Ghias, Naveed Rashid, Yousra Sarfaraz, Muhammad Umer Naeem, Syed Murtaza Raza Kazmi, Muneeb Khalid, Khairulnissa Ajani, Qamar Riaz, Mehmood Riaz, Rozina Roshan, Sibtain Ahmed.

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
