## [Editor Report · Decision Letter 0]

7 Oct 2025

Evidence-Based Action Plan for Integrating Artificial Intelligence in an Academic Medical Centre-A Multidisciplinary Approach

PLOS ONE

Dear Dr. Jafri,

Thank you for submitting your manuscript to PLOS ONE. After careful consideration, we feel that it has merit but does not fully meet PLOS ONE’s publication criteria as it currently stands. Therefore, we invite you to submit a revised version of the manuscript that addresses the points raised during the review process.

To progress toward peer review, the manuscript must be revised to include a detailed and explicit discussion of study's limitations, potential biases, and the justification or calculation for the sample size. Furthermore, to comply with the journal's reporting standards, the COREQ checklist must be fully completed and submitted alongside the revised manuscript. Once these changes are made and the required checklist is provided, we will assess the manuscript's suitability for progression to the peer-review process.

We look forward to receiving your revised manuscript.

Kind regards,

Syed Hani Abidi

Academic Editor

PLOS ONE

Journal Requirements:

2. We note that your Data Availability Statement is currently as follows: “All relevant data are within the manuscript and its Supporting Information files.”

---

## [Author Response · Author response to Decision Letter 1]

28 Oct 2025

October 2025

To the Editor-in-Chief

PLOS ONE

Subject: Rebuttal letter for article titled- “Evidence-Based Action Plan for Integrating Artificial Intelligence in an Academic Medical Centre-A Multidisciplinary Approach”

Dear Sir,

Thank you for your feedback and guidance. We have addressed all concerns raised point by point below.

• Thank you for submitting your manuscript to PLOS ONE. After careful consideration, we feel that it has merit but does not fully meet PLOS ONE’s publication criteria as it currently stands. Therefore, we invite you to submit a revised version of the manuscript that addresses the points raised during the review process. To progress toward peer review, the manuscript must be revised to include a detailed and explicit discussion of study's limitations, potential biases, and the justification or calculation for the sample size.

Response: Study now includes limitations, potential biases, and the justification or calculation for the sample size.

• Furthermore, to comply with the journal's reporting standards, the COREQ checklist must be fully completed and submitted alongside the revised manuscript. Once these changes are made and the required checklist is provided, we will assess the manuscript's suitability for progression to the peer-review process.

Response: the COREQ checklist has been submitted now.

• When submitting your revision, we need you to address these additional requirements. Please ensure that your manuscript meets PLOS ONE's style requirements, including those for file naming. The PLOS ONE style templates can be found at https://journals.plos.org/plosone/s/file?id=wjVg/PLOSOne_formatting_sample_main_body.pdf and https://journals.plos.org/plosone/s/file?id=ba62/PLOSOne_formatting_sample_title_authors_affiliations.pdf

Response: We have revised the manuscript according to the PLOS ONE style requirements and ensured that all files are renamed and formatted in line with the provided templates including the title page.

• We note that your Data Availability Statement is currently as follows: “All relevant data are within the manuscript and its Supporting Information files.” Please confirm at this time whether or not your submission contains all raw data required to replicate the results of your study. Authors must share the “minimal data set” for their submission. PLOS defines the minimal data set to consist of the data required to replicate all study findings reported in the article, as well as related metadata and methods (https://journals.plos.org/plosone/s/data-availability#loc-minimal-data-set-definition). For example, authors should submit the following data:- The values behind the means, standard deviations and other measures reported;- The values used to build graphs;- The points extracted from images for analysis. Authors do not need to submit their entire data set if only a portion of the data was used in the reported study. If your submission does not contain these data, please either upload them as Supporting Information files or deposit them to a stable, public repository and provide us with the relevant URLs, DOIs, or accession numbers. For a list of recommended repositories, please see https://journals.plos.org/plosone/s/recommended-repositories. If there are ethical or legal restrictions on sharing a de-identified data set, please explain them in detail (e.g., data contain potentially sensitive information, data are owned by a third-party organization, etc.) and who has imposed them (e.g., an ethics committee). Please also provide contact information for a data access committee, ethics committee, or other institutional body to which data requests may be sent. If data are owned by a third party, please indicate how others may request data access.

Response: We have already shared the themes, even codes and analysis of the qualitative data. The full data cannot be shared due to confidentiality restrictions. However all relevant data supporting the findings including coding, themes etc are described in the paper. The study can be replicated using the details provided in the manuscript.

Response: Not applicable

Regards,

Corresponding Author:

Dr Lena Jafri

---

## [Decision Letter · Decision Letter 1]

26 Dec 2025

Dear Dr. Jafri,

Specifically, the authors should revise the abstract to provide a concise scientific summary that includes a clear justification for the chosen methods, a brief description of the analytic procedures, and concrete, validated outcomes. Additionally, restructure the introduction to clearly articulate a specific research gap and validate the selection of the SWITCH model over alternative frameworks.

In the methods section, adopt a coherent qualitative design by clearly defining whether you are using a grounded-theory approach or alternate method. Provide a justification for the prospective sample size and address aspects of researcher reflexivity, along with triangulation through explicit procedures.

Finally, deepen the results and action plan sections by incorporating richer, analytically driven themes, including illustrative quotations and a validated, prioritized implementation framework. The discussion must also critically reflect on methodological limitations and ethical considerations around AI usage, power dynamics, structural constraints.

plosone@plos.org. . . . A letter that responds to each point raised by the academic editor and reviewer(s). You should upload this letter as a separate file labeled 'Response to Reviewers'.A marked-up copy of your manuscript that highlights changes made to the original version. You should upload this as a separate file labeled 'Revised Manuscript with Track Changes'.An unmarked version of your revised paper without tracked changes. You should upload this as a separate file labeled 'Manuscript'.

We look forward to receiving your revised manuscript.

Kind regards,

Syed Hani Abidi

Academic Editor

PLOS One

Journal Requirements:

Reviewers' comments:

Reviewer's Responses to Questions

**Comments to the Author**

Reviewer #1: (No Response)

Reviewer #2: (No Response)

2. Is the manuscript technically sound, and do the data support the conclusions?

Reviewer #1: Yes

Reviewer #2: Partly

3. Has the statistical analysis been performed appropriately and rigorously?

Reviewer #1: Yes

Reviewer #2: Yes

4. Have the authors made all data underlying the findings in their manuscript fully available?

Reviewer #1: Yes

Reviewer #2: Yes

5. Is the manuscript presented in an intelligible fashion and written in standard English?

Reviewer #1: No

Reviewer #2: Yes

Reviewer #1: Abstract-The abstract is largely descriptive and reads more like a project summary than a scientific abstract. Critical elements are missing, including a clear justification for methodological choices, explicit analytic procedures, and concrete outcomes beyond thematic labels. Claims of providing a “structured strategic action plan” are not supported by evidence of validation, implementation, or evaluation.

Introduction-The introduction provides a generic overview of AI in healthcare and change management theories but lacks a sharply articulated research gap. The rationale for selecting the SWITCH model over other established frameworks remains underdeveloped and largely asserted rather than justified. The novelty of the study is unclear, particularly given the abundance of conceptual papers on AI integration and leadership in healthcare.

Methods-This is the weakest section of the manuscript.

• The study is labeled as “qualitative cross-sectional,” yet the design blends interviews, workshops, and consensus meetings without a coherent qualitative methodological framework.

• Claims of using grounded theory are not supported by methodological rigor (e.g., no constant comparison, no theory generation, no memoing).

• Sample size justification is post-hoc and conceptually weak; reliance on a pilot of two interviews to justify adequacy is not defensible.

• The role of researchers (many of whom are institutional insiders and co-authors) raises significant concerns regarding reflexivity, positionality, and social desirability bias, which are insufficiently addressed.

• Triangulation is asserted but not demonstrated analytically.

Results-The results are largely descriptive and repetitive.

• Themes are broad, unsurprising, and closely aligned with existing literature, offering limited analytical depth.

• Coding appears largely confirmatory, reinforcing pre-existing assumptions about AI rather than generating new insights.

• Quotations are sparse and selectively illustrative rather than analytically rich.

• Tables function more as checklists than as vehicles for theory building or interpretive synthesis.

Action Plan (Phase III)-The action plan is extensive but essentially prescriptive and aspirational.

• It resembles a policy or institutional guideline rather than a research output.

• No prioritization framework, feasibility testing, stakeholder validation beyond authors, or implementation metrics are provided.

• The plan is not empirically evaluated, piloted, or compared against alternatives, limiting its scholarly contribution.

Discussion-The discussion reiterates results without critically engaging with methodological limitations or alternative interpretations.

• Claims of scalability and transferability are speculative and not supported by comparative data.

• The discussion conflates relevance with rigor; importance of topic does not compensate for methodological weakness.

• Engagement with critical AI ethics, power dynamics, and structural constraints in LMIC contexts remains superficial.

Limitation-Although a limitations section is present, it is largely formulaic and understated.

• Key issues such as insider bias, lack of independent validation, and absence of outcome assessment are not adequately acknowledged.

• The implications of these limitations for the credibility of the action plan are not discussed.

Reviewer #2: The paper has a relevant and timely focus on AI inclusion in medical education and research, and action plan using the SWITCH model), which could be attractive if methodological rigor, clarity, and positioning are strengthened. The core issues now are precision and depth in the qualitative methodology, sharper articulation of novelty, and substantial tightening of language and structure. These will enhance the article and will benefit wider audience.

.

Reviewer #1: No

Reviewer #2: **Yes:**Srinivasa Rao BollaSrinivasa Rao BollaSrinivasa Rao BollaSrinivasa Rao Bolla

NAAS will assess whether your figures meet our technical requirements by comparing each figure against our figure specifications.<button aria-busy="false" aria-describedby="tooltip-750" class="button button-for-icon button-small button-ghost-weak lumo-no-copy" style="box-sizing: inherit; margin: 0px; padding: 0px; border-width: 1px; border-style: solid; border-color: rgba(0, 0, 0, 0); border-image: none 100% / 1 / 0 stretch; background-image: initial; background-position: initial; background-size: initial; background-repeat: initial; background-attachment: initial; background-origin: initial; background-clip: initial; font: inherit; color: rgb(12, 12, 20); appearance: none; cursor: pointer; vertical-align: middle; padding-block: 4.28571px; padding-inline: 4.28571px; border-radius: 8px; outline: unset; transition: 0.15s cubic-bezier(0.22, 1, 0.36, 1), background-position; --button-default-background-color: transparent; --button-hover-background-color: rgba(186, 190, 199, 0.2); --button-active-background-color: rgba(186, 190, 199, 0.3); --button-default-text-color: #0c0c14; --button-hover-text-color: #0c0c14; --button-active-text-color: #0c0c14; --padding-block: .3571428571em; --padding-inline: .3571428571em; quotes: "“" "”" "‘" "’"; min-block-size: 0px; min-inline-size: 0px; scrollbar-width: thin; scrollbar-color: rgba(0, 0, 0, 0) rgba(0, 0, 0, 0);" type="button">

</button>

---

## [Author Response · Author response to Decision Letter 2]

23 Jan 2026

Jan 2026

To

The Editor-in-Chief

PLOS ONE

Subject: Response to reviewers’ comments for article titled “Evidence-Based Action Plan for Integrating Artificial Intelligence in an Academic Medical Centre-A Multidisciplinary Approach”

Dear Sir,

We have revised the manuscript in accordance with the academic editor’s and reviewers’ suggestions. A point-by-point response is provided below, along with both a tracked-changes version and a clean revised manuscript submitted on the PLOS One portal.

1. Specifically, the authors should revise the abstract to provide a concise scientific summary that includes a clear justification for the chosen methods, a brief description of the analytic procedures, and concrete, validated outcomes. Additionally, restructure the introduction to clearly articulate a specific research gap and validate the selection of the SWITCH model over alternative frameworks.

Response: Abstract has been revised as per suggestions. We clarify that the aim of this study was exploratory and developmental, focusing on the co-creation of a practical framework rather than the validation of outcomes. We have revised the manuscript to clarify the exploratory nature of the study, explicitly describe the analytic process, articulating a clearer research gap, and reason behind choosing SWITCH over other models of change for devising the action plan.

2. In the methods section, adopt a coherent qualitative design by clearly defining whether you are using a grounded-theory approach or alternate method. Provide a justification for the prospective sample size and address aspects of researcher reflexivity, along with triangulation through explicit procedures.

Response: We agree with reviewers. In this study, the grounded theory approach has been removed in favor of an exploratory research design that used thematic analysis. This method aimed to uncover unknown patterns and themes from interview and workshop discussion data, employing an inductive approach to ensure findings were data-driven and allowed themes to emerge organically from participants' responses, aligning with the exploratory nature of the study. By employing the concepts of information power and theme saturation to clearly justify the anticipated sample size, we have now improved the Methods section. Additionally, we have included a clear reflexivity statement and described the triangulation techniques used, such as data source and analyst triangulation.

3. Finally, deepen the results and action plan sections by incorporating richer, analytically driven themes, including illustrative quotations and a validated, prioritized implementation framework. The discussion must also critically reflect on methodological limitations and ethical considerations around AI usage, power dynamics, structural constraints.

Response: Results have been edited, added more quotations, tried writing it analytically with interpretations where needed. Since this was beyond the scope of the study, the action plan was not validated; however, the SWITCH model of change framework was employed, and the results were integrated into the SWITCH framework. The Discussion has been revised focusing on literature review, ethics and limitations of the study.

4. Reviewer #1: Abstract-The abstract is largely descriptive and reads more like a project summary than a scientific abstract. Critical elements are missing, including a clear justification for methodological choices, explicit analytic procedures, and concrete outcomes beyond thematic labels. Claims of providing a “structured strategic action plan” are not supported by evidence of validation, implementation, or evaluation.

Response: The abstract has been revised as per these suggestions of writing it scientifically. We wish to clarify that the manuscript does not claim formal validation or implementation of the action plan. Rather, it presents a conceptual, structured action plan developed from qualitative findings and aligned with the SWITCH model, intended to guide institutions in AI integration. We have revised the text to make this distinction clearer and avoid any potential misunderstanding.

5. Introduction-The introduction provides a generic overview of AI in healthcare and change management theories but lacks a sharply articulated research gap. The rationale for selecting the SWITCH model over other established frameworks remains underdeveloped and largely asserted rather than justified. The novelty of the study is unclear, particularly given the abundance of conceptual papers on AI integration and leadership in healthcare.

Response: The Introduction has been strengthened to clearly articulate the existing gaps, novelty and the rationale for conducting this study. The SWITCH model was selected for its focus on practical behaviour change among healthcare stakeholders, rather than just organizational restructuring. It uniquely incorporates cognitive, emotional, and environmental factors, suitable for the study's multi-phase design involving interviews, discussions, and workshops over six months. This flexibility allowed its application across various study stages and stakeholder groups. The manuscript has been revised to more clearly justify the selection of the SWITCH model (in the introduction).

6. Methods-This is the weakest section of the manuscript. The study is labeled as “qualitative cross-sectional,” yet the design blends interviews, workshops, and consensus meetings without a coherent qualitative methodological framework.

Response: Thank you for this comment, we have revised the methods section and agree and the study design has been modified too.

7. Methods-Claims of using grounded theory are not supported by methodological rigor (e.g., no constant comparison, no theory generation, no memoing).

Response: In this study, the grounded theory approach has been removed in favor of an exploratory research design that used thematic analysis. This method aimed to uncover unknown patterns and themes from interview and workshop discussion data, employing an inductive approach to ensure findings were data-driven and allowed themes to emerge organically from participants' responses, aligning with the exploratory nature of the study. We have revised the Methods section to remove references to grounded theory, as the study did not aim to generate a formal theory.

8. Methods-Sample size justification is post-hoc and conceptually weak; reliance on a pilot of two interviews to justify adequacy is not defensible.

Response: We respectfully clarify that the sample size was not determined post hoc. The proposed sample size of interviews was specified a priori and approved by the institutional ethics review committee before data collection commenced. We agree, however, that our original explanation relied too heavily on the pilot interviews. We have therefore revised the Methods section to remove the pilot based justification and now explicitly ground the sample size rationale in established qualitative principles, including purposive sampling of information-rich participants and anticipated thematic saturation.

9. Methods-The role of researchers (many of whom are institutional insiders and co-authors) raises significant concerns regarding reflexivity, positionality, and social desirability bias, which are insufficiently addressed. Triangulation is asserted but not demonstrated analytically.

Response: Agree, this has been addressed in the methods section now. We have updated the Methods section to specifically consider researcher positionality, including prior experience and institutional familiarity. Confidentiality was guaranteed to participants, and open-ended questions were used in neutral, non-evaluative interviews to promote candid observations. Throughout the process of gathering + analyzing data, researchers kept written reflections to track how their viewpoints might have affected interpretations. To increase credibility and lower the possibility of social desirability bias, results were also triangulated across participant views and cross-checked with pertinent documents whenever feasible.

10. Results-The results are largely descriptive and repetitive.

Response: This was a helpful comment. We agree that earlier version of results were too descriptive. Results section has been revised by the authors and we have tried to remove repetition and tried to strengthen analytical depth.

11. Results-Themes are broad, unsurprising, and closely aligned with existing literature, offering limited analytical depth.

Response: We recognize that multiple themes presented align with existing literature. The themes were developed inductively and later discussed with workshop and meeting participants. In response to the reviewer's critique, we have enhanced the Results section by providing analytic commentary within each theme, elucidating the interconnections among themes to enhance interpretive depth.

12. Results-Coding appears largely confirmatory, reinforcing pre-existing assumptions about AI rather than generating new insights.

Response: We thank the reviewer for this comment. We respectfully clarify that the analysis was not intended to be confirmatory. Any alignment with existing literature would therefore have been unintentional. An inductive coding approach was employed, deriving codes from participants’ accounts instead of predefined theories. Acknowledging the initial presentation's shortcomings, the Methods and Results sections were revised to better articulate the inductive coding process and enhance the analytic commentary, emphasizing how participant perspectives influenced and expanded existing understandings of AI implementation.

13. Results-Quotations are sparse and selectively illustrative rather than analytically rich.

Response: We have revised the Results section as per this helpful suggestion to ensure that quotations are used analytically rather than illustratively. Pre or post quotation analysis have been added where appropriate. More quotes have been added. In the initial version quotations were used selectively to avoid overly descriptive results. But now we understand how relevant quotations can be added.

14. Results-Tables function more as checklists than as vehicles for theory building or interpretive synthesis.

Response: We agree with the reviewer and this is an important observation. We have tried to edit the tables as much as we could to strengthen this aspect of the paper.

15. Action Plan (Phase III)-The action plan is extensive but essentially prescriptive and aspirational. It resembles a policy or institutional guideline rather than a research output.

Response: We appreciate the reviewer’s recognition of the action plan’s scope and practical relevance. The action plan was analytically derived which is not coming through. We have clarified its analytic intent to ensure it is read as a research output rather than a prescriptive guideline.

16. Action Plan (Phase III)-No prioritization framework, feasibility testing, stakeholder validation beyond authors, or implementation metrics are provided. The plan is not empirically evaluated, piloted, or compared against alternatives, limiting its scholarly contribution.

Response: We appreciate the reviewer’s insightful critique and agree that empirical evaluation, feasibility testing, stakeholder validation and comparative analysis would strengthen the work. We wish to clarify that the primary aim of this manuscript is to propose a conceptual and integrative framework, rather than to report empirical findings. Please note, data collection (both in interviews and also in workshop) included many organizational leaders and decision-makers, who function as key stakeholders in the context of this study. This informally does provide stakeholder validation beyond the authorship team.

17. Discussion-The discussion reiterates results without critically engaging with methodological limitations or alternative interpretations. Claims of scalability and transferability are speculative and not supported by comparative data. The discussion conflates relevance with rigor; importance of topic does not compensate for methodological weakness. Engagement with critical AI ethics, power dynamics, and structural constraints in LMIC contexts remains superficial.

Response: Discussion has been rewritten taking account of literature, limitations and avoiding repetition of results. Moving beyond a summary of results, we now critically evaluate our findings against existing literature. While the importance of the topic remains a driver, we have reinforced the study's rigor by being more transparent about our methodological choices and the validity of our conclusions. Please note the findings from the current study offer potential insights into how healthcare institutes may integrate AI based on study findings. These insights are exploratory and provide a foundation for future empirical and comparative studies. Therefore, claims about scalability and transferability are presented as potential insights rather than confirmed outcomes. Further studies with larger samples and comparative data are needed to validate these aspects.

18. Limitation-Although a limitations section is present, it is largely formulaic and understated. Key issues such as insider bias, lack of independent validation, and absence of outcome assessment are not adequately acknowledged. The implications of these limitations for the credibility of the action plan are not discussed.

Response: Limitations section has been written and we have tried to address the potential biases, validations and outcome assessment need.

19. Reviewer #2: The paper has a relevant and timely focus on AI inclusion in medical education and research, and action plan using the SWITCH model), which could be attractive if methodological rigor, clarity, and positioning are strengthened. The core issues now are precision and depth in the qualitative methodology, sharper articulation of novelty, and substantial tightening of language and structure. These will enhance the article and will benefit wider audience.

Response: Thank you for the comments. As per reviewers’ and editor’s suggestions methods and results have been modified bringing clarity, analytical interpretation and relevance. We have tried to improve writing and structure of the manuscript too.

---

## [Decision Letter · Decision Letter 2]

10 Mar 2026

Dear Dr. Jafri,

We look forward to receiving your revised manuscript.

Kind regards,

Syed Hani Abidi

Academic Editor

PLOS One

Journal Requirements:

Reviewers' comments:

Reviewer's Responses to Questions

**Comments to the Author**

Reviewer #1: (No Response)

Reviewer #2: All comments have been addressed

2. Is the manuscript technically sound, and do the data support the conclusions?

Reviewer #1: Yes

Reviewer #2: Yes

3. Has the statistical analysis been performed appropriately and rigorously?

Reviewer #1: N/A

Reviewer #2: Yes

4. Have the authors made all data underlying the findings in their manuscript fully available?

Reviewer #1: Yes

Reviewer #2: Yes

5. Is the manuscript presented in an intelligible fashion and written in standard English?

Reviewer #1: Yes

Reviewer #2: No

Reviewer #1: The manuscript addresses a timely and contextually important issue; however, before it can be considered for publication, several substantive concerns require further attention. In particular,

1. The study should explicitly and consistently define its methodological orientation (e.g., exploratory qualitative study using reflexive thematic analysis) and ensure alignment between research questions, data collection methods (interviews, workshop, meetings), and analytic procedures.

2. Provide a clearer step-by-step account of how codes were generated, how themes were refined, and how decisions were made during analysis. A visual analytic map or thematic development table would enhance transparency.

3. Move beyond descriptive summaries of what participants said. For each theme, include interpretive commentary that explains why these patterns matter, how they relate to existing literature, and what tensions or contradictions emerged.

4. Integrate quotations analytically rather than illustratively. Each quote should be followed by interpretation that demonstrates how it advances conceptual understanding rather than merely exemplifying a theme.

5. Given the insider status of several authors, include a more explicit reflexivity statement detailing positionality, power dynamics, and how potential bias was managed throughout data collection and analysis.

6. Specify how interview findings, workshop discussions, and planning meetings were compared, reconciled, and synthesized into the final action plan. The analytic integration process requires clearer articulation.

7. Provide a more critical comparison with alternative change-management frameworks, explaining precisely what conceptual gap the SWITCH model fills in this context.

8. Clearly distinguish the action plan from a policy guideline by articulating the theoretical insights derived from the data. Consider presenting a prioritization matrix or conceptual model that demonstrates analytic synthesis rather than listing components.

9. Explicitly acknowledge insider bias, absence of external validation, lack of pilot testing, and absence of outcome evaluation, and discuss how these limitations affect the credibility and applicability of the proposed framework.

Reviewer #2: Major strengths

• Timely, practice‑oriented topic in an LMIC

• Focus on AI integration in a large academic medical centre in Pakistan addresses a real gap, as most AI‑change papers are conceptual or from HICs.

• Multi‑source qualitative design

• Three phases (leadership interviews, multidisciplinary workshop, then action‑plan development) create a coherent, developmental trajectory from perceptions → co‑design → framework.

• Inclusion of leaders, faculty, and students increases the breadth of perspectives despite a single‑site limitation.

• Explicit change‑management lens (SWITCH)

• Clear mapping of findings to “Direct the Rider / Motivate the Elephant / Shape the Path” gives the action plan a structured and theoretically anchored form.

Recommended modifications

• In Abstract and Methods, standardize to one primary descriptor, e.g.: “We conducted an exploratory qualitative study using reflexive thematic analysis of interviews and a co‑design workshop, followed by collaborative development of an action plan.”

At the start of Methods, add a short “Study design and phases” paragraph that clearly lists:

Phase I: Individual semi‑structured interviews with institutional leaders and faculty.

Phase II: Co‑design workshop with multidisciplinary participants, using SWITCH as a sensitizing framework.

Phase III: Collaborative development of a SWITCH‑aligned action plan based on Phases I–II.

• In the discussion, there are many repetitions about the SWITCH model's benefits; keep it once at the beginning.

• Break the limitations into bullet points or paragraphs

• Check the conflict of interest statement and correct

• Check for spelling mistakes (pliot, emperical, effectiveness, particpants)

.

Reviewer #1: No

Reviewer #2: **Yes:**Srinivasa Rao BollaSrinivasa Rao BollaSrinivasa Rao BollaSrinivasa Rao Bolla

---

## [Author Response · Author response to Decision Letter 3]

18 Mar 2026

2026/03/16

To the Editor PLOS One

Subject: Response to Reviewers’ Comments

Dear Editor,

Thank you for sharing the reviewers’ comments and as per their suggestions we have revised our manuscript. All comments are responded below against each point and manuscript has been edited accordingly. We have submitted our revised manuscript in 3 separate files:

1. This document detailing the changes made

2. a copy of the manuscript with track-changes

3. a clean copy of the revised manuscript

• Reviewer #1: The manuscript addresses a timely and contextually important issue; however, before it can be considered for publication, several substantive concerns require further attention. In particular,1. The study should explicitly and consistently define its methodological orientation (e.g., exploratory qualitative study using reflexive thematic analysis) and ensure alignment between research questions, data collection methods (interviews, workshop, meetings), and analytic procedures. Response: We thank the reviewer for the opportunity to further clarify our methodological alignment. We have now explicitly labelled the study as an exploratory qualitative study using reflexive thematic analysis in the opening of the Methods section. Now the study objectives align with the research question, methodology and data analysis.

• 2. Provide a clearer step-by-step account of how codes were generated, how themes were refined, and how decisions were made during analysis. A visual analytic map or thematic development table would enhance transparency. Response: The methodology section has been revised to explicitly detail the systematic steps of thematic analysis, ensuring greater clarity and transparency in the research process. We have added a Visual Analytic Map (Figure 1) to the Methodology section to enhance transparency. This figure illustrates the iterative process of manual coding and clarifies how the workshop acted as a validation bridge between the initial inductive analysis and the final deductive mapping to the Switch Model.

• 3. Move beyond descriptive summaries of what participants said. For each theme, include interpretive commentary that explains why these patterns matter, how they relate to existing literature, and what tensions or contradictions emerged. Response: Agreed, in each theme we have added interpretive commentaries where possible.

• 4. Integrate quotations analytically rather than illustratively. Each quote should be followed by interpretation that demonstrates how it advances conceptual understanding rather than merely exemplifying a theme. Response: agree all the quotations are now integrated analytically rather than illustratively.

• 5. Given the insider status of several authors, include a more explicit reflexivity statement detailing positionality, power dynamics, and how potential bias was managed throughout data collection and analysis. Response: We acknowledge our insider status within the hospital. To manage potential bias, data collection was framed as collaborative problem solving rather than a top down evaluation, reducing perceived hierarchies. Trustworthiness was further ensured through methodological triangulation, comparing interview data with workshop findings, and guaranteeing anonymity to encourage candid participation and mitigate social desirability bias. We have tried to explain this in the methods section.

• 6. Specify how interview findings, workshop discussions, and planning meetings were compared, reconciled, and synthesized into the final action plan. The analytic integration process requires clearer articulation. Response: This has been modifed and rewritten in the methodology.

• 7. Provide a more critical comparison with alternative change-management frameworks, explaining precisely what conceptual gap the SWITCH model fills in this context. Response: Thank you for pointing this out. We have now added the critical comparison of models of change now in the introduction section.

• 8. Clearly distinguish the action plan from a policy guideline by articulating the theoretical insights derived from the data. Consider presenting a prioritization matrix or conceptual model that demonstrates analytic synthesis rather than listing components. Response: We agree that a distinction between a policy guideline and a research-derived action plan is vital. We have utilized the Switch framework (Rider, Elephant, and Path check table 2) not merely as a categorization tool, but as a conceptual model for analytic synthesis. This framework allows us to demonstrate how the qualitative themes translate into specific behavioural and environmental changes. The action plan is intentionally structured for practical utility to ensure knowledge translation which is a key goal of applied qualitative research. While it may resemble a policy guideline in its clarity, it is distinguished by its inductive origin. To emphasize this, we have added explicit links between the proposed actions and the emergent themes from the data. All edits are in track changes.

• 9. Explicitly acknowledge insider bias, absence of external validation, lack of pilot testing, and absence of outcome evaluation, and discuss how these limitations affect the credibility and applicability of the proposed framework. Response: We thank the reviewer for this critical observation. We agree that the use of a specific model and the current stage of the framework's development present significant limitations to its immediate applicability. We have revised the Limitations section to explicitly discuss how the insider bias, lack of pilot testing and how the SWITCH model may have introduced interpretive bias and how the absence of external validation and outcome evaluation necessitates caution when generalizing these findings.

• Reviewer #2: Major strengths • Timely, practice oriented topic in an LMIC • Focus on AI integration in a large academic medical centre in Pakistan addresses a real gap,as most AI change papers are conceptual or from HICs. • Multi source qualitative design• Three phases (leadership interviews, multidisciplinary workshop, then action plan development) create a coherent, developmental trajectory from perceptions → co design → framework. • Inclusion of leaders, faculty, and students increases the breadth of perspectives despite a single site limitation.• Explicit change management lens (SWITCH)• Clear mapping of findings to “Direct the Rider / Motivate the Elephant / Shape the Path” gives the action plan a structured and theoretically anchored form. Response: thank you for these positive encouraging comments.

• Recommended modifications• In Abstract and Methods, standardize to one primary descriptor, e.g.: “We conducted an exploratory qualitative study using reflexive thematic analysis of interviews and a co design workshop, followed by collaborative development of an action plan.”At the start of Methods, add a short “Study design and phases” paragraph that clearly lists:Phase I: Individual semi structured interviews with institutional leaders and faculty.Phase II: Co design workshop with multidisciplinary participants, using SWITCH as a sensitizing framework.Phase III: Collaborative development of a SWITCH aligned action plan based on Phases I–II. Response: Thank you for this suggestion. We have standardized the primary descriptor of the study design throughout the Abstract and Methods sections to ensure clarity and consistency. As requested, a "Study design and phases" subsection has been added to the beginning of the Methods section.

• • In the discussion, there are many repetitions about the SWITCH model's benefits; keep it once at the beginning. Response: Thankyou we have now edited the discussion and benefits of SWITCH have been merged together.

• • Break the limitations into bullet points or paragraphs. Response: Agree, we have now written the study limitations as bullet points for clarity.

• • Check the conflict of interest statement and correct. Response: Thank you for pointing out this error. The statement has been rewritten.

• • Check for spelling mistakes (pliot, emperical, effectiveness, particpants). Response: thank you we have now removed all grammatical errors and spelling mistakes.

---

## [Decision Letter · Decision Letter 3]

22 Mar 2026

Evidence-Based Action Plan for Integrating Artificial Intelligence in an Academic Medical Centre-A Multidisciplinary Approach

PONE-D-25-46411R3

Dear Dr. Jafri,

We’re pleased to inform you that your manuscript has been judged scientifically suitable for publication and will be formally accepted for publication once it meets all outstanding technical requirements.

Kind regards,

Syed Hani Abidi

Academic Editor

PLOS One

Additional Editor Comments (optional):

Reviewers' comments:

Reviewer's Responses to Questions

**Comments to the Author**

Reviewer #1: All comments have been addressed

2. Is the manuscript technically sound, and do the data support the conclusions?

Reviewer #1: Yes

3. Has the statistical analysis been performed appropriately and rigorously?

Reviewer #1: N/A

4. Have the authors made all data underlying the findings in their manuscript fully available?

Reviewer #1: Yes

5. Is the manuscript presented in an intelligible fashion and written in standard English?

Reviewer #1: Yes

Reviewer #1: No furthere comments.The paper address all the comments.

The manuscript is now technically sound, ethically robust, and provides a valuable contribution to the field of AI integration in healthcare. I find no further concerns regarding dual publication or research ethics. All previous queries have been successfully addressed, and I recommend the paper for immediate acceptance

.

Reviewer #1: No

---

## [Editor Report · Acceptance letter]

PONE-D-25-46411R3

PLOS One

Dear Dr. Jafri,

I'm pleased to inform you that your manuscript has been deemed suitable for publication in PLOS One. Congratulations! Your manuscript is now being handed over to our production team.

Kind regards,

on behalf of

Dr. Syed Hani Abidi

Academic Editor

PLOS One